# Bridging a Century-Old Problem: The Pathophysiology and Molecular Mechanisms of HA Filler-Induced Vascular Occlusion (FIVO)—Implications for Therapeutic Interventions

**DOI:** 10.3390/molecules27175398

**Published:** 2022-08-24

**Authors:** Danny J. Soares

**Affiliations:** 1American Foundation for Aesthetic Medicine (AFFAM), Fruitland Park, FL 34731, USA; drsoares@plasticsurgeryvip.com; 2College of Medicine, University of Central Florida, Orlando, FL 32827, USA

**Keywords:** hyaluronic acid, dermal fillers, hyaluronidase, urokinase, fibrinolytic therapy, skin necrosis, retinal injury, cerebrovascular stroke, pulmonary embolism, angiosome

## Abstract

Biocompatible hyaluronic acid (HA, hyaluronan) gel implants have altered the therapeutic landscape of surgery and medicine, fostering an array of innovative products that include viscosurgical aids, synovial supplements, and drug-eluting nanomaterials. However, it is perhaps the explosive growth in the cosmetic applications of injectable dermal fillers that has captured the brightest spotlight, emerging as the dominant modality in plastic surgery and aesthetic medicine. The popularity surge with which injectable HA fillers have risen to *in vogue* status has also brought a concomitant increase in the incidence of once-rare iatrogenic vaso-occlusive injuries ranging from disfiguring facial skin necrosis to disabling neuro-ophthalmological sequelae. As our understanding of the pathophysiology of these injuries has evolved, supplemented by more than a century of astute observations, the formulation of novel therapeutic and preventative strategies has permitted the amelioration of this burdensome complication. In this special issue article, we review the relevant mechanisms underlying HA filler-induced vascular occlusion (FIVO), with particular emphasis on the rheo-mechanical aspects of vascular blockade; the thromboembolic potential of HA mixtures; and the tissue-specific ischemic susceptibility of microvascular networks, which leads to underperfusion, hypoxia, and ultimate injury. In addition, recent therapeutic advances and novel considerations on the prevention and management of muco-cutaneous and neuro-ophthalmological complications are examined.

## 1. Introduction

The advent of chemically crosslinked hyaluronic acid (HA, hyaluronan) gel derivatives has heralded a new era in the biomedical arena, yielding novel therapeutic applications in the fields of surgery, regenerative medicine, and pharmacology [1,2]. However, it is perhaps the utilization of hydrocolloidal HA-based dermal fillers in aesthetic medicine that has witnessed the most significant growth in the last decade, commanding the largest share of the global cosmetic market in 2022 [3,4,5]. The explosion in the use of these injectable HA gels has brought a concomitant rise in the incidence of once-rare ischemic tissue infarcts stemming from the accidental intravascular injection of gel boluses, resulting in filler-induced vascular occlusion (FIVO). With our growing understanding of the injury mechanisms of FIVO, novel therapeutic interventions have become possible. In this article, we review the current state of knowledge relating to the pathophysiology of this iatrogenic complication, with important implications for treatment and prevention.

### 1.1. Historical Perspective

The use of injectable compounds in plastic surgery originated in the late 1890s with the use of soft hydrocarbon prostheses in the correction of facial deformities [6,7]. At the time, the practice involved the injection of a soft mixture of paraffin and vaseline through a large bore needle, followed by the immediate sculpting, cooling, and fixation of the treated areas (Figure 1) [8]. Although the material was later found to be prone to migration and chronic inflammatory reactions, this novel application of injectables in aesthetic medicine gave birth to a powerful non-surgical treatment modality, positively expanding the horizons of the specialty [9,10,11].

Despite initial acclaim and optimism, paraffin injections struck a historically ominous chord when early occurrences of sudden blindness, stroke, and pulmonary emboli—occurring immediately following injection—began to be described circa 1903 [12]. Though initially baffled, contemporary surgeons at the time eventually surmised that the inciting cause of such devastating events stemmed from the inadvertent intravascular injection and dissemination of soft paraffin into the ophthalmic, cerebral, and pulmonary vasculatures, resulting in tissue infarction [13]. These reports described the very first known instances of FIVO, which, together with paraffin’s predisposition toward disfiguring granulomas, culminated in the eventual cessation of paraffin use by the late 1920s [14].

Over the ensuing century, the post-paraffin evolution of cosmetic injectables progressed through other suboptimal alloplastic compounds—most notably liquid silicone in the 1950s—prior to eventually settling on the use of homologous biocompatible materials like collagen (1980s) and hyaluronan (2000s) [15,16,17]. These agents, though of limited longevity, offered an unmatched safety profile and a low incidence of adverse reactions or complications. With the FDA approval of the first cosmetic HA dermal filler (Restylane^®^, Galderma, Fort Worth, TX, USA) in 2002, and the subsequent flourishing of aesthetic injectables, hyaluronan solidified its hegemony as the leading cosmetic injection material of the new millennium [18]. However, as filler treatments ascended to the forefront of plastic surgery, the resurgence of vascular occlusion events soon followed, portending the continuation of a century-old medical saga in the 21st century.

### 1.2. Modern Parallels

In the two decades since the introduction of HA injectable gels, dermal fillers have witnessed a surge in popularity, with sales booming by 700% between 2005 and 2020 [19,20]. As of 2022, the global market for cosmetic fillers registered a value of $5.31 billion and a growth forecast of 65% by 2030 [21]. In the United States alone, ~3.4 million dermal filler procedures are performed yearly, with HA fillers comprising nearly 80% of filler products [22]. This increased demand for cosmetic injectables stems from the growing acceptance of plastic surgery procedures across all age and gender groups, as well as the increased accessibility and marketing, as well as decreased cost of products, creating an increasingly competitive market that is likely to favor further adoption.

Despite the positive influence of dermal fillers on the specialty, their widespread utilization has brought the concomitant rise in the incidence of ischemic injuries stemming from FIVO. Beleznay et al. reported nearly 50 cases of filler-induced blindness within a period of approximately 3 years between 2015 and 2018, standing in sharp contrast to the 98 published instances of such occurrences in the preceding century [23,24]. Similarly, the incidence of filler-induced ischemic skin injuries has shown a 30-fold increase between 2000 and 2020, while that of filler-induced stroke has risen by 300% (Figure 2) [25,26]. In 2015, in response to this well-documented hazard, the FDA ordered the re-labelling of all filler products to include a warning statement on the risk of vascular occlusion [27].

### 1.3. Clinical Scope of the Problem

The clinical scope of injuries resulting from FIVO is ample owing to the numerous arterial and venous conduits of the face, which enable the dissemination of embolized fragments to local and distant targets. Consequently, published reports of FIVO injuries have encompassed vision loss, ophthalmoplegia, cerebral infarction, skin necrosis, pulmonary embolism, cerebral sinus thrombosis, and even death [28,29,30,31,32,33,34]. Nevertheless, the exact magnitude of the problem has been difficult to quantify owing to intrinsic limitations in the U.S. Federal Manufacturer and User Facility Device Experience (MAUDE) database, a passive reporting system prone to under-reporting and information deficiencies [35].

A recent 2021 FDA panel on the risks associated with intravascular injection of dermal fillers disclosed a total of 470 vascular adverse events reported within a 5-year period between 2015 and 2020. Of those, at least 91% occurred in the face, with the perioral, nasal, and nasolabial sites of injection comprising the majority (68%) of such incidents (Figure 3). Notably, nearly 20% of reported injuries were complicated by vision-related sequelae, with 85% of patients incurring persistent deficits [36]. The potential for iatrogenic blindness with FIVO raises the specter of not only debilitating injuries, but also the risk of practice-ending litigation, posing bilateral costly ramifications to patients and practitioners alike [37].

The cosmetic deformities that can result from FIVO-associated skin necrosis, which carry the potential for facial disfigurement, are also significant. A recent systematic review of 243 photographic reports of FIVO resulting in facial skin ischemia revealed that the facial and ophthalmic arterial angiosomes were frequently involved, comprising 58% and 48% of injuries, respectively. In particular, the glabellar, nasal, and upper lip regions, all of which share critical aesthetic significance, were the facial zones most frequently affected by ischemic skin loss [25]. The damage caused by facial skin necrosis carries substantial negative financial and quality of life repercussions that impose permanent burdens on affected patients, reinforcing the urgent need for greater preventative and therapeutic initiatives [38,39].

## 2. Hyaluronan Biophysiology and Aesthetic Applications

The popularity of HA as an injectable material has placed it at the forefront of FIVO thanks to its widespread use in aesthetic medicine. First isolated from the bovine ocular vitreous and described by Meyer and Palmer in 1934, HA is a biomacromolecule with important physiological functions, remarkable versatility, and numerous applications in medicine [40,41]. Since the elucidation of its chemical structure in 1954, HA has been routinely employed in wound dressings, ophthalmological viscosurgical aids, and synovial viscosupplementation products [42,43]. Following the advent of genetically modified bacterial fermentation processes that enabled the large-scale manufacturing of HA derivatives, the number of available HA products has grown substantially and now includes a wide array of applications that include the correction of facial rhytids, volume depletion, and contour loss [44,45,46,47].

### 2.1. Basic Biochemical Properties and Function

Hyaluronan is a ubiquitous heteropolysaccharide consisting of alternating disaccharide units composed of β-1,4-D-glucuronic acid and β-1,3-N-acetyl-D-glucosamine (Figure 4) [48]. Biochemically, the HA macromolecule is a stable non-sulfated, non-branched glycosaminoglycan with important structural, hygroscopic, and signaling roles within the extracellular matrix (ECM) [49,50]. The human body contains approximately 15 g of hyaluronan—capable of binding 1000× its weight in water—with nearly half of all HA present in the skin and featuring a resident half-life of only 24 h [51]. In vivo, HA exists as a highly hydrated, polyanionic molecule ranging in weight from 100 kDa in serum to 7000 kDa in the vitreous humor [52,53]. Because of their charged nature, hyaluronan molecules display significant intra- and inter-molecular hydrogen bonds, providing secondary and tertiary structural rigidity, forming anti-parallel helical duplexes that can assemble into infinitely large meshworks, imparting this molecule its viscoelastic properties (Figure 5) [54,55,56].

The biological functions performed by hyaluronan in living tissues are numerous. Structurally, HA’s vast hygroscopic meshwork imparts a hydrating, space-filling quality to the extracellular milieu that facilitates the cellular and biochemical activities of tissues. The osmotic and viscoelastic characteristics of HA also enhance the soft tissue turgor, pliability, and resiliency of the dermis, conferring a youthful quality to the skin [57]. Physiologically, hyaluronan serves as an important mediator and signaling molecule in wound healing, interacting directly with CD44 and modulating epidermal growth factor receptor (EGFR) and transforming growth factor-β1 (TGFβ1) receptor activity [58,59,60,61]. Furthermore, individual HA molecules also possess signaling properties that are dependent on molecular weight. Short (<20 kDa) and medium (20–500 kDa) molecules demonstrate pro-angiogenic, immunostimulatory actions, while high-molecular weight (HMW) HA (>500 kDa) shows the opposite tendency toward an immunodepressive effect [62].

### 2.2. Physiological HA Biosynthesis and Degradation

In eukaryotes, the synthesis of hyaluronan is governed by a family of glycosyl-transferases named HA synthases (HASs), which convert two distinct uridine diphosphate (UDP)-sugar precursors (UDP-glucuronic acid and UDP-N-acetylglucosamine) into HMW HA [63]. In humans, the production of hyaluronan relies on three different isoenzymes—HAS1, HAS2, and HAS3—all of which are transmembrane proteins that polymerize HA from cytosolic substrates directly into the extra-cellular environment. HAS2 carries the most significant functional role, based on the lethality of HAS2 knockouts, and is responsible for most HA production in tissues [64,65]. The regulation of HA synthesis appears to be dependent on a variety of signaling factors associated with inflammatory and wound healing mediators, including transforming growth (TGF-β), insulin-like growth factor (IGF), fibroblast growth factor (FGF), and prostaglandins [66,67].

The degradation of HA is executed by endogenous hyaluronidases, endo-beta-N-acetylhexosaminidases that hydrolyze endogenous and exogenous HA. In humans, six homologous genes encoding for hyaluronidases have historically been described: HYAL-1 through HYAL-4, PHYAL-1, and SPAM-1 [68]. Of these, HYAL-1 and 2 are by far the most active in human tissues, with HYAL-2 existing as a free, unbound enzyme responsible for the breakdown of most extracellular HA into intermediate-sized polysaccharides and HYAL-1 serving as the membrane-bound form in lysosomes capable of further degrading endocytosed HA into its constituent monosaccharides [69]. The intravascular half-life of hyaluronidases is short, spanning only 2–3 min because of the presence of plasma glycoprotein inhibitors [70]. In tissues, hyaluronidases have a more prolonged activity, remaining active for 24–48 h; nevertheless, some murine studies have shown a complete drop in activity after 3–6 h [71,72]. For this reason, hyaluronidase re-dosing every 1–6 h has been recommended in cases of FIVO [73]. Additionally, other HA degradation pathways have been described recently that also bear significance. Specifically, the clearance of intravascular HA appears to rely on endocytosis via HARE (HA receptor for endocytosis)-mediated binding by the sinusoidal endothelial cells in the liver and spleen [74]. In addition, a new membrane-bound protein, HYBID (hyaluronan binding protein involved in hyaluronan depolymerization), has been identified and shown to play a significant role in HA turnover in the skin. HYBID appears to participate in an endocytosis-mediated breakdown of extracellular HA and its senescence-related dysfunction has been implicated in the pathophysiology of skin elastosis and aging [75].

### 2.3. Commercial HA Synthesis and Reversal Agents

The commercial manufacturing of raw HA precursors constitutes a multi-billion-dollar industry subserving a wide range of cosmetic, dietary, and pharmaceutical-grade products [76]. The production of injectable HA gels requires HMW (>1000 kDa) raw HA that must adhere to established safety and purity standards [77]. Bacterial fermentation is currently the largest source of HMW HA, having largely supplanted the more-costly, lower-yield, animal-derived preparations from rooster combs and bovine vitreous. Specifically, the bacterium *S. equis*, subspecies *zoopidemicus*—a gram-positive, capsule-forming Lancefield type C streptococcus—is the most-commonly employed strain used in HA production [78]. Compared with animal-derived sources, bacterial-based methods demonstrate enhanced-purity, higher molecular weight, lower immunogenicity, and greater product yield [79]. Nonetheless, new heterologous recombinant expression systems harnessing *B. subtilis* and even cell-free in vitro production systems employing a soluble form of Class II HA synthases show promise in further achieving enhanced yield rates, higher molecular weight chains, and diminished polydispersity [80,81,82]. Prior to serving as the raw ingredient in the fabrication of dermal fillers, HMW HA (500–2500 kDa) derived from bacterial cultures is precipitated in isopropanol before undergoing a robust purification process that removes endogenous toxin remnants from the streptococcal source [83].

The rising demand for HA gel fillers has spurred a proportionate growth in the manufacturing of hyaluronidase mixtures to be used in the rapid degradation and reversal of injected hyaluronan products. In the United States, several brands have received FDA approval for the enhancement in tissue hydration and drug diffusion, though their use as HA reversal agents currently still remains off-label. The human recombinant form (Hylenex^®^, Halozyme Therapeutics Inc., San Diego, CA, USA) is currently the most favored among aesthetic clinicians because of its low risk of hypersensitivity reactions and higher potency, though bovine (Amphadase^®^, Amphastar Pharmaceuticals Inc., Rancho Cucamonga, CA, USA) and ovine (Vitrase^®^, ISTA Pharmaceuticals Inc., Irvine, CA, USA) varieties are available [84,85]. These products are supplied as small, single-use 1–2 mL vials containing limited quantities (150–200 USP units/mL) that are significantly smaller than the high doses (500–1500 U, q 1–6 h) currently advocated in the management of FIVO injuries [86,87,88].

### 2.4. Rheological Properties and Particle Size

Modern HA-based dermal fillers consist of 1,4-butanediol diglycidyl ether (BDDE)-crosslinked HMW hyaluronan molecules (Figure 6) suspended in a variable amount of a carrier solution of uncrosslinked HA [89]. Though the chemical features of each gel vary according to brand-specific proprietary crosslinking technologies, they demonstrate a degree of modification (MoD) that ranges from 1 to 10%, a molecular weight of 100–600 kDa, and HA concentrations (HAC) ranging from 15 to 24 mg/mL [90,91]. This relatively high content of long-chain, crosslinked hyaluronan imparts a viscoelastic solid behavior to these products—with energy dissipation factors (tan δ values) of less than 1—that allows them to resist shear and compressive deformation in vivo [92]. Fine-tuning of the MoD and HAC enables manufacturers to formulate fillers with different shear elastic moduli (G’) and dynamic strength/stretch characteristics that enhance the product’s suitability for superficial versus deep injection [93]. As of 2022, nearly two dozen individual brands/sub-brands of HA fillers exist in the U.S. market, with the number expected to continue to increase into the future [22].

The macroparticle profile of HA dermal fillers varies with each product but bears significance considering the embolic potential of gel fragments in FIVO. Restylane-L^®^ and Restylane Silk^®^/Lyft^®^ (Galderma, Fort Worth, TX, USA), colloquially known as “bi-phasic” fillers, reportedly undergo a proprietary sieving/milling process that generates uniformly sized microparticles immersed in a free HA carrier gel, roughly 25% by volume [94,95]. The particle sizes for these fillers range from 50 to 220 μm (Restylane Silk^®^), 330 to 430 μm (Restylane-L^®^), and 750 to 1200 μm (Restylane Lyft^®^) [96]. However, the majority of current HA fillers—including Juvéderm^®^ brands (Allergan Aesthetics, Irvine, CA, USA); Restylane Defyne^®^, Refyne^®^ Kysse^®^, and Contour^®^ (Galderma, Fort Worth, TX, USA); RHA^®^ 2, 3, 4 and Redensity^®^ (Revance, Nashville, TN, USA); Revanesse Versa^®^ (Prollenium, Raleigh, NC, USA); and Belotero Balance^®^ (Merz, Raleigh, NC, USA)—are considered “monophasic”, representing a continuous, uniform gel matrix that is nonsieved and non-particulate [97,98]. Nevertheless, when dispersed in an aqueous solution, monophasic fillers separate into fragments that also range between 50 and 1200 μm in size [99,100]. Thus, HA fillers are naturally prone to fragmentation and dispersion when exposed to high shear forces during injection, potentially generating embolizable macro- and microparticles.

The viscoelastic properties of HA hydrogels, though beneficial in their wrinkle-correcting prowess, present unique challenges when placed intravascularly, with several rheological properties potentially influencing their behavior within vessels (Table 1). Specifically, the ability of a gel to take up water, known as the swelling factor, can directly impact the gel particle volume as it hydrates intraluminally. Currently available fillers exhibit swelling factors that range between 100 and 700% by volume, increasing their occlusive potential [101,102]. In addition, cohesivity, or the capacity of a gel to resist fragmentation and dispersal, may bear significance in the intravascular behavior of dermal fillers and their response to reversal agents [103]. Monophasic gels, because of their nonparticulate nature, are more cohesive and less likely to break up in buffered solutions or when exposed to hyaluronidase than bi-phasic fillers [104]. The cohesiveness of a product may affect its tendency to fragment, micro-embolize, and resist rapid degradation with reversal agents [99,105]. Finally, the shear elastic modulus (G’), which characterizes a gel’s stiffness and ability to resist shear deformation, may impact a product’s tendency to occlude or compress a vessel. High G’ fillers could resist blood flow more effectively than softer, low G’ gels; in addition, stiffer gels may possess a greater ability to externally compress vessels and further reduce blood flow [106,107]. Nonetheless, these rheological properties of fillers can be disrupted by simple extrusion through small-bore devices, such as 30 g needles, indicating that the properties of fillers intravascularly may differ from those of filler in the syringe [108]. Additional research is currently necessary on the behavior of HA hydrogels in different intravascular environments.

## 3. Pathophysiology of HA-Mediated Vascular Occlusion

The mechanism of injury in FIVO represents a multifactorial cascade of events that includes vasocannulation, vasoinoculation, vasodissemination, and vasoocclusion, resulting in tissue underperfusion and ischemia (Figure 7). Although arterial compression by extravascular filler boluses is recognized as a valid alternative etiology of FIVO, such a mechanism is rare outside of instances involving surgerized tissues with a tenuous blood supply or those nourished by end-arteries with absent cross-perfusion from adjacent angiosomes. In normal tissues, compression is likely to result in diminished blood flow, inducing pallor and decreased capillary refill, but without complete loss of perfusion or tissue necrosis [109]. In mouse and rabbit FIVO models, arterial compression by filler could not be induced and did not result in any prolonged period of significant underperfusion or ischemia [110,111]. In humans, single-point obstructions of facial vessels are easily overcome by a vastly redundant blood supply from adjacent angiosomes, via true anastomoses, which dilate post filler-placement [112]. Even in instances of bilateral external carotid artery (ECA) ligation, historically performed for intractable epistaxis, compensatory flow from the internal carotid artery (ICA) system has been sufficient to avoid tissue necrosis [113,114,115]. Therefore, outside of compression of distal arterial branches in patients with abnormally underperfused tissues, the intravascular model of FIVO is likely to be the most prevalent. In this section, we review the pathophysiology of this intravascular model and outline preventative and therapeutic opportunities that exist along each phase of injury.

### 3.1. Vaso-Cannulation

The accidental penetration of a blood vessel during active filler injection represents the triggering event that immediately precedes FIVO [116,117]. Blunt-tipped microcannulas and sharp hypodermic needles, both routinely employed in aesthetic injectable treatments, can potentially perforate vessel walls and initiate intraluminal inoculation of filler. Microcannulas, because of their flexible shaft and closed tip configuration, have generally been regarded as more prudent and are endorsed by multiple consensus panels on injection safety [118,119]. Compared with needles, microcannulas cause less tissue trauma, pain, edema, and bruising, and can achieve a more reliable plane-specific placement of filler [120,121,122,123]. When employed by specialized practitioners with knowledge of vascular anatomy, microcannula use has demonstrated a significantly lower risk of FIVO compared with needle injections [124]. Nevertheless, higher gauge microcannulas (27 g and up) have vasopenetration forces equivalent to those of needles (1–2 kg⋅m/s^2^), suggesting that their safety advantage is limited or nonexistent in many treatment applications [125,126].

The likelihood of vessel cannulation also depends on the force necessary to advance a needle/cannula subcutaneously. Histologically, the central facial tissues of the perioral and perinasal regions feature thicker, more fibrous interconnections between the muscular aponeurosis and the skin (Figure 8) [127]. In those regions, the superficial musculoaponeurotic system (SMAS) displays a type II morphology, which lacks the soft intervening adipose layer and smooth gliding plane that characterize the type I tissues of the lateral face. The more restrained nature of vessels within type II SMAS and the tissue’s greater resistance to cannulation mean that even blunt devices, such as microcannulas, may easily perforate through vascular structures, as evidenced by multiple illustrative case reports [128,129,130,131,132].

Because of the central role of vaso-cannulation in the mechanism of FIVO, preventative steps have been suggested that help reduce the odds of intraluminal placement. Pre-injection aspiration, in which the plunger of the syringe is pulled back to generate a negative-pressure screening for visible blood, has been controversial at best [133,134]. Although a positive aspiration may equate with a higher likelihood of intravascular placement, the unsteady movements transmitted to the tip during aspiration may arguably negate any preventative value from this maneuver beyond that of false reassurance. Furthermore, the routine use of a stationary injection technique, itself necessitated by the practice of aspiration, mandates the bolus-based placement of filler that could magnify the potential degree of injury [135]. In addition, filler-primed needle hubs, which characterize the majority of injections, carry a high incidence of false-negative aspirations, creating a misleading perception of safety [136]. Nonetheless, aspiration with both air- and saline-primed needles has demonstrated excellent sensitivity in both in vitro and in vivo testing in medium-sized arteries, though its incorporation into clinical practice could prove cumbersome and time-consuming [137].

The use of Doppler ultrasound (US) has recently grown in acceptance as a means of identifying aberrant/anomalous vascular structures prior to dermal filler injection and avoiding accidental vaso-cannulation [138,139]. Proponents of US advocate its routine use predominantly in high-risk areas—such as the ophthalmic artery-supplied territories of the forehead, glabella, and nasal dorsum —where it can be employed for both arterial mapping and ultrasound-guided filling [140,141,142]. US may also prove useful in cases of FIVO, helping to identify sites of vascular occlusion and assisting with the precise delivery of reversal agents [143,144]. However, the significant cost, skill training, and time-consuming aspects of US use in clinical practice have thus far limited its widespread adoption [145]. As portable US technology continues to evolve, this real-time imaging modality is likely to play a more pivotal role in the prevention and management of vascular occlusion injuries.

### 3.2. Vaso-Inoculation

In FIVO, the intraluminal inoculation of a blood vessel represents a brief, but significant event in which a variable amount of filler undergoes a pressurized delivery into the intravascular milieu. Conceptually, different mechanisms of inoculation exist depending on the positioning of the needle/cannula tip relative to the lumen of the vessel: intraluminal versus extraluminal (Figure 9) [146]. Intraluminal positioning results in a more efficient delivery of inoculum than extraluminal placement, in which the material must work back into a vessel along the track created by the needle/cannula. Given the blind nature of filler injections and the substantial needle movements that occur during filling, especially in light of the small diameters of most facial vessels, a combination of extraluminal and intraluminal mechanisms is likely to occur.

The injection of a viscoelastic hydrogel into an arterial lumen requires a favorable pressure gradient, such that sufficient force must be applied to the plunger to overcome the mean arterial pressure [147]. In a fresh cadaver head perfusion model, Cho et al. found that an average injection pressure of 166 mm Hg was sufficient to inoculate the supratrochlear artery (STrA) with Juvéderm Ultra and retrogradely fill the ophthalmic artery back to the takeoff point of the retinal artery [148]. Because of the pseudoplastic nature of HA dermal fillers, ejection pressures required to overcome the gel’s yield stress—upon which gel fillers begin to flow smoothly through the narrow lumen of a needle—are well above mean arterial values [149]. Lee et al. demonstrated that even soft fillers with a low shear elastic modulus (G’) and high tan delta extruded from a 25 g needle at a pressure of 580 mm Hg, with maximal extrusion pressures as high as 6500 mm Hg for high G’ fillers injected through 30 g needles [150]. Given these elevated ejection pressures, any cannulation of a vessel that occurs during active filler injection is likely to result in vaso-inoculation, underscoring the need for better preventative steps that help minimize the former.

In FIVO, the volume of inoculum strongly influences the extent of the resultant injury. Small, microscopic amounts of HA gel are likely to be immediately dispersed and embolized distally, whereas large boluses may form an advancing plug that completely occludes a segment of the vascular lumen [151]. Because even the largest extracranial facial vessels feature narrow luminal sizes ranging between 1 and 2 mm in diameter, the volumes of filler required to completely occlude a large segment of an artery are small [152,153,154,155,156]. Cho et al. and Kahn et al. revealed that as little as 0.05–0.1 mL can extensively occlude the 5 cm segment of ophthalmic artery located between its supratrochlear and retinal branches [148,157]. Given the rapid, high-pressure extrusion dynamics characteristic of filler injections against the relatively small volume/low resistance characteristics of facial vessels, even brief inoculation periods can deliver sufficient filler to cause obstruction [148]. In addition, because of the high viscosity and cohesivity of many HA fillers—with drop weights ranging from 15 to 50 mg—even small amounts of HA filler may potentially achieve a complete blockage of facial vessels [103]. Thus, the injection force and speed strongly influence the volume of intravascular inoculum, and thus also the extent of injury. Consequently, several safety consensus panels have recommended that practitioners employ a slow, low-pressure injection technique during treatment and limit filler bolus sizes to less than 0.1 mL [158,159].

### 3.3. Vaso-Dissemination

The intravascular behavior of HA gels is incompletely understood, but several animal studies and post hoc histopathological analyses have offered insight into the patterns of dissemination of dermal fillers [160,161,162]. The type of blockade induced by HA gels seemingly depends on the volume of inoculum as well as the rheological properties and particle size profile of the offending product. Small amounts of a low-viscosity filler injected into a large-caliber artery likely result in complete distal dispersion, while large volumes of a high-viscosity filler are more likely to produce an obstructive plug. Nie et al. proved this to be true by showing a higher rate of total, irreversible vessel occlusion with HA (Restylane^®^) than compared with injections of a lower viscosity, small-particle polymethylmethacrylate (PMMA) filler (Artecoll^®^; Hafod BV, Rotterdam, The Netherlands) in the rabbit-ear model [151]. However, a comparative rabbit ear study employing equivalent amounts of different HA (Restylane^®^, Juvederm^®^ Ultra, and Belotero^®^) and non-HA (Radiesse^®^; Merz USA, Greensboro, NC, USA) products did not identify differences in the extent of ischemic injury or necrosis, suggesting that most HA fillers are more alike than they are dissimilar in their intravascular dispersal [111].

The vaso-disseminating behavior of a gel inoculum is also dictated by whether the affected vessel is arterial or venous in nature. Arterial inoculations are probably responsible for most of the local injuries occurring in the head and neck, while venous cannulations likely account for the majority of distant injuries (Table 2), although exceptions may occur owing to the presence of arteriovenous shunts, which have been described within the head and neck [163]. Intravenously placed facial filler can remain stationary—inciting a locoregional thrombophlebitic reaction that may intensify up to cerebral sinus thrombosis—or may undergo cardiopetal embolization to ultimately lodge within the pulmonary arterial tree, resulting in a pulmonary embolism [164,165]. Furthermore, venous occlusions can contribute to tissue ischemia even in cases stemming from arterial-cannulation because of the presence of arteriovenous shunting, contributing to superimposed congestive venous failure [166]. Ultimately, however, whether they are local, distant, venous, or arterial, the ischemic injury mechanism incited by FIVO bears an arterio-occlusive etiology. As a result, in this section, the intra-arterial mechanisms of filler dispersal will be the predominant focus of discussion.

Conceptually, four different patterns of vaso-dissemination of HA filler are possible, depending on the volume of inoculum and properties of the filler (Figure 10). 

In type I, the filler is completely dispersed and embolized into the distal vascular tree, resulting in the complete elimination of the gel material from within the main arterial lumen. Type I vaso-dissemination is likely to occur with low-volume injections of low-viscosity/non-cohesive fillers into large-caliber vessels (Appendix A). Conversely, type II dissemination represents the formation of an isolated proximally obstructive plug at the site of injection within the main arterial lumen without any significant distal dispersion. A single-point occlusion can result in varying outcomes, from a minimal perfusion defect to an extensive injury, depending on the availability of distal anastomotic collateral circulation (Figure 11). Type II dissemination requires a higher viscosity/cohesivity filler injected in sufficiently large volumes to form a plug. Finally, type III and IV patterns feature a combination of both distal dispersion and proximal luminal obstruction and are likely the most common patterns, occurring with soft, moderately dispersible filler materials injected in sufficiently large volumes. Type IV patterns differ from type III by having the additional retrograde advancement of filler, which can result in the embolization and blockage of further upstream branches (Appendix A).

The extent of filler dissemination, and thus the potential severity of injury that may be incited by the inoculation of an artery, depends on the effective size of its accessible arteriovascular territory (AAT). The AAT represents the segments of all affected angiosomes that are susceptible to filler occlusion, i.e., the conglomerate of all arterial branches located distal to the most proximal site of occlusion into which filler may naturally spread, causing obstruction (Figure 12) [167]. As described by Taylor and others, the size of the functional angiosome, and hence the AAT, depends on the presence of true anastomoses and the behavior of choke vessels [168,169]. True anastomoses function as permanent connections between adjacent arterial territories or across arteriovenous shunts, effectively increasing the size of the AAT, often dramatically [170]. In contrast, choke vessels tend to restrict access to adjacent arterial territories in a reflexive manner that is thought to be protective, reducing the AAT [171]. However, the constriction of choke vessels probably contributes to decreased tissue perfusion in cases of FIVO, just as it does in surgical flaps, potentially compounding the degree of ischemia [172,173]. Thus, the dissemination of intravascular filler depends on multiple variables, many of which are patient-, tissue-, and filler-specific, resulting in differing patterns of occlusion.

### 3.4. Vaso-Occlusion

#### 3.4.1. HA Bolus-Mediated Occlusion

In FIVO, the occlusion of a vascular network represents the final outcome of a dynamic battle between the pressurized intraluminal environment and the resiliency of a gel plug, which is gradually deformed, degraded, fragmented, and dispersed before settling into an equilibrium (Appendix A). Once the dissemination of intraluminal filler is completed, the resulting distribution of gel fragments determines the extent of the occlusive injury. In several animal and human models of FIVO, this distribution typically consists of proximal intraluminal plugs with distally impacted filler microemboli (1–1000 μm), consistent with type III/IV vaso-dissemination (Figure 13) [111,174,175].

These distal emboli can fully or partially obstruct any component of the arterial tree, including small arterial branches (~400–1000 μm), subcutaneous arterioles (~100–400 μm), terminal (dermal) arterioles (15–100 μm), and capillaries (5–15 μm) [148,177,178,179,180,181]. Ultimately, the ischemic insult of FIVO results from the underperfusion of end-organ capillary beds and the amount of filler required to do so depends on the functional configuration of the affected angiosome; specifically, it depends on the presence of true anastomoses and choke vessels [182,183].

True anastomotic connections enhance the resiliency of tissue perfusion through the existence of secondary flow paths that can circumvent occlusive plugs. In addition, true anastomoses also expand the size of the AAT, increasing the volume of filler necessary to significantly disrupt tissue perfusion. Nonetheless, the presence of true anastomoses widens the reach of filler emboli, allowing for the distant spread of filler. In contrast, tissues that lack true anastomoses depend on the patency of choke vessels, which serve as functional gatekeepers governing the flow of blood between adjacent vascular territories [184]. In FIVO, choke vessels may reflexively vasoconstrict, limiting the ability of filler particles to spread to adjacent angiosomes [171]. However, this protective effect of choke vessels also cuts off potentially vital collateral perfusion to tissues, increasing the risk of ischemia. The presence of arteriovenous connections (AVCs) also influences the pattern of occlusion and dysperfusion in FIVO. AVCs may be useful in clearing filler particles from arterial lumens into the venous system; however, they can also hinder the adequate perfusion of tissues [163]. This shunting capability, recently described, explains the presence of large gel particles in venules following intra-arterial injections in the rabbit ear model [166].

The clinical presentation of FIVO varies with the vascular configuration of the affected tissues. For instance, filler-induced injuries stemming from type III dissemination in the supratrochlear artery (STrA)—in which the artery distal to the injury site is occluded—show a multitude of ischemic skin patterns (Figure 14) [185,186,187,188,189,190,191,192,193]. These variations reflect the presence of existing anastomoses between the STrA and the dorsal nasal artery, paracentral artery, supraorbital artery, and distal branches of the anterior division of the superficial temporal artery [194]. Consequently, patterns of cutaneous necrosis range from insignificant to extensive. In contrast, injuries arising from type IV dissemination of filler within the STrA frequently present with acute irreversible vision loss (Figure 15) [195]. The severity of injury reflects the end-arterial nature of the retinal circulation, in which the central retinal artery (CRA) is the sole nourishing conduit for the inner retina—specifically the macula, responsible for central, high-resolution, color vision—in approximately 70% of individuals (Figure 16) [196,197,198]. The absence of anastomoses renders the CRA one of the most susceptible vascular systems in the entire human body, though the existence of a cilioretinal branch in 30% of the population may limit the severity of CRA occlusion (CRAO) injuries [199,200].

The pattern of vascular occlusion may carry significance for the rapid and effective delivery of reversal agents. Because type III/IV dissemination patterns have been most frequently implicated in human and animal instances of FIVO, both intravascular and extravascular delivery of hyaluronidase are likely to be of benefit. Multiple in vitro and in vivo animal studies involving the rat femoral artery and rabbit ear models have demonstrated that extravascular hyaluronidase is effective in penetrating through the arterial wall to induce the sufficient degradation of intravascular HA, averting necrosis [175,201,202,203]. Extravascular hyaluronidase was found to be most effective when given early (within 4 h), at high doses, and distributed over multiple treatment sessions [174,204]. Similarly, intravascular hyaluronidase was effective in the treatment of FIVO in the rabbit ear model, especially if injected within the first 4 h post injury, as well as in humans for the treatment of ischemic skin injuries [205,206].

In contrast, extravascular reversal therapy has encountered repeated failure and shown limited benefit in cases of retinal artery occlusion. Specifically, outside of some rare reports of success, most studies evaluating the effectiveness of retro-bulbar delivery of extravascular hyaluronidase in CRAO have failed to elicit a significant improvement or yielded only limited benefit [207,208,209,210]. This may be because of the technically challenging nature of retro-bulbar injections, with imprecise delivery of the enzyme into the vicinity of the CRA, or may be in part influenced by impaired diffusion across the arterial wall in the retro-orbital compartment. Fortunately, intra-arterial therapy has demonstrated a modest, but encouraging degree of success in the reversal of CRAO, especially when intervention is administered early and is combined with thrombolytic therapy [211,212]. Nonetheless, therapeutic success rates are still suboptimal, as evidenced by the poor outcomes reported in multiple animal and human interventional studies, likely due to delays in therapy [181,213,214]. Furthermore, cases of iatrogenic cerebral infarct occurring during super-selective angiography underscore the risks of intra-arterial therapy [215]. Consequently, the use of high-dose intravenous hyaluronidase is currently being explored as a rapid, more accessible alternative to intra-arterial therapy, with some promising outcomes [216,217]. Given the short half-life of hyaluronidase in plasma (2–3 min), a high-dose, continuous infusion may be necessary to achieve sufficient reversal activity at distant sites of occlusion, potentially increasing the adverse effects of such systemic therapy [216,218,219].

#### 3.4.2. Thrombus-Mediated Occlusion

The vaso-occlusive process initiated by an intraluminal bolus of HA gel induces a pro-thrombotic state that can further extend the size of an obstructive plug. This effect appears to be triggered by hemostasis and a direct HA–platelet interaction capable of initiating platelet aggregation [216]. Several studies in the murine epigastric and femoral artery models have confirmed this effect, in which the formation of an early platelet-rich “white thrombus” within the gel plug subsequently extends beyond the bolus to form a fibrin-rich “red thrombus” (Figure 17) [160,175]. The immediacy of this thrombotic pathway suggests an inflammatory etiology initiated by the HA bolus rather than anoxic endothelial injury. Furthermore, the white nature of the early thrombus implies a direct role for platelet interaction with intravascular HA [220]. Indeed, multiple studies have described a pro-inflammatory, platelet-activating effect of HA initiated by platelet-derived hyaluronidase 2 and leukocyte binding [221,222,223]. Furthermore, in a comparative study by Nie et al., thrombus formation was evident in Restylane-inoculated rabbit ears, but absent in PMMA-inoculated animals, indicating that HA specifically induces a thrombotic response [151]. This is additionally supported by the finding that platelet-free human serum fails to show any clot-inducing properties when mixed with dermal fillers in vitro [224].

The thrombogenicity of vasoinoculated HA gels bears significance in the pathophysiology of FIVO and is made evident by treatment recommendations featuring anti-platelet and thrombolytic therapies [86]. Although oral anti-platelet therapy with 325 mg aspirin has traditionally been advocated, its exact benefit has not yet been quantified. Thrombolytic therapy, on the other hand, has recently been shown to improve outcomes in FIVO. Multiple studies in the rat epigastric and femoral vein HA-occlusion models have shown a significant flap survival benefit in animals treated with dual therapy—featuring thrombolytic agents (urokinase or alteplase) combined with hyaluronidase—compared with hyaluronidase monotherapy [160,175,216]. Furthermore, this benefit was present with both the subcutaneous and intravenous administration routes, which circumvented the arterial injury risk associated with intra-arterial interventions [225]. In humans, dual intra-arterial therapy has demonstrated some benefit, performing favorably when compared with monotherapy [226]. Zhang et al. demonstrated an improvement in visual perception in 42% of patients presenting with CRAO stemming from FIVO who were treated with combined intra-arterial therapy, performing comparatively better than monotherapy [211]. Nonetheless, the prompt initiation of treatment remains the most essential component of successful therapy and is likely responsible for the limited therapeutic success rates witnessed beyond the 3–4.5 h post-injury treatment window [227,228,229].

### 3.5. Tissue Ischemia

The terminal stage of the injurious mechanism of FIVO is prolonged tissue ischemia. An ischemic state is defined physiologically as the absence of a minimum degree of tissue oxygenation necessary for basal metabolic function and cell survival [230,231,232]. At the biochemical level, ischemia deprives the cell of its ATPase-dependent ion transport and volume regulatory mechanisms, resulting in organelle dysfunction and eventual lysis of plasma membranes [233]. Ischemia is said to be reversible when prompt restoration of perfusion allows for the unaltered functional survival of tissues and irreversible when permanent injury has been inflicted [234]. The concept of ischemic penumbra is often employed to describe underperfused tissues for which irreversible injury is assured given sufficiently prolonged ischemic times, particularly with neuronal tissues [235]. The identification of tissue ischemia in FIVO is typically established surrogately on the basis of clinical manifestations of injury, loss of function, or through angiography [230]. Given the range of tissues that may be affected by FIVO, clinicians must remain vigilant for a wide variety of signs and symptoms.

The susceptibility of tissues to ischemic injury depends on the metabolic activity and survival demands of their cellular constituents. The ischemic tolerance of tissues relates to the maximum period of ischemia that an organ can tolerate prior to suffering irreversible injury, varying from minutes to days depending on tissue type [236]. The ischemic injury reflects both the deleterious effects of anoxic damage as well as the superimposed ischemia-reperfusion injury (IRI) brought on by the return of tissue circulation and oxygenation. The substantial harm caused by IRI is incited by a reactive oxygen species (ROS)-mediated endothelial injury, which subsequently triggers a prothrombotic state that culminates in capillary-occlusion and further anoxic injury [233,237]. The ischemic tolerance time of the human retina is short, with irreversible injury occurring as early as within 90 min of occlusion and possibly even sooner [238,239]. The limited survivability of ischemic retinal tissue reflects its extremely high basal metabolic rate and unique nutrient demands, analogous to those of other neuronal tissues [240]. Likewise, the human brain is unable to tolerate ischemic times extending beyond 1.5–4.5 h, while the skin may survive 12–24 h of ischemia [241,242].

#### 3.5.1. Ischemic Skin Injury

Ischemic skin injuries represent the most common complication of FIVO, encompassing approximately 80% of clinical manifestations [36]. Unaddressed, underperfused skin will eventually progress to skin necrosis, resulting in tissue loss and permanent scarring.

On clinical examination, FIVO-associated skin injuries frequently present with disproportionate pain and skin discoloration in 80% and 70% of cases, respectively [25]. The appearance of compromised skin typically progresses through multiple stages over the first 7–10 days post-injury (Figure 18). Initially, within the first several hours post-occlusion, pallor and delayed capillary refill become evident, followed by livedoid skin changes. Livedo reticularis—a dusky, violaceous, and mottled discoloration exhibited by the skin in the first 72 h post-injury—represents an accumulation of deoxygenated blood in the post-capillary venules of the skin as a result of arterial blockade [243]. As the injury progresses, the integrity and function of the skin become compromised—enabling the gradual deterioration of its barrier properties with increased bacterial bioburden—before eventually undergoing coagulative necrosis with escharification [86].

In the face, the superficial distribution of the skin injury follows a non-random, but variable pattern governed by the configuration of its arterial vascular network. The blood supply to the face counts on generous contributions from branches of the internal carotid artery (ICA) and external carotid artery (ECA) (Figure 19). 

Specifically, main dermal perforasomes emanating from the ophthalmic artery (ICA), facial artery (ECA), internal maxillary artery (ECA), and distal external carotid/superficial temporal artery supply the entirety of the mucocutaneous surfaces of the face and anterior oronasal cavities [244,245]. Each perforasome represents a cutaneous territory fed by one or more arterial skin perforators interconnected by direct (true) subcutaneous anastomoses and indirect (choke) sub-dermal anastomoses [246,247,248]. In the human face, the cutaneous blood supply shows some evidence of compartmentalization arranged into a motif that approximates the superficial fat pads originally described by Rohrich and Pessa, which explains the recurring clinical patterns of injury in the face [249,250,251]. This vascular configuration forms the basis of the newly proposed **F.O.E.M.** (**F**acial, **O**phthalmic, distal **E**xternal carotid, and internal **M**axillary) scoring system (Figure 20) and grading classification (Table 3) of severity of FIVO-associated skin injuries [25]. However, because of the intrinsic variability in the nature of arterial branching and anastomotic connections, FIVO facial skin injuries demonstrate some clinical diversity as well.

Facial skin ischemia threatens the integrity, function, and appearance of the face and, as such, represents an aesthetic emergency [252]. Multiple therapeutic avenues have been endorsed in the management of FIVO-associated skin injuries, though some still lack sufficient supporting evidence [87,88]. Reversal therapy employing hyaluronidase, with and without thrombolytics, remains the first-line intervention with the greatest potential impact. Hyaluronidase therapy should be initiated promptly (ideally within the first 4 h post-injury), employ high doses, and be performed over multiple sessions spaced regularly over the first 24–48 h [119,146]. Treatment should immediately target underperfused regions demonstrating delayed capillary refill as well as any suspected sites of main arterial occlusion [150,174,206].

Hyperbaric oxygen therapy (HBOT), though not yet shown objectively to impact tissue survival in FIVO, has been associated with favorable outcomes in multiple case reports/series and is supported by a long history of therapeutic benefit in flap-based surgeries [253,254,255]. HBOT increases tissue oxygenation through increased plasma oxygen tension, enhancing the diffusion reach of oxygen in partially perfused tissues, improving the ischemic tolerance of tissues, and promoting angiogenesis [256]. This hyperoxygenation effect appears to persist for several hours following each treatment session [257]. However, in instances of severe occlusion, HBOT alone is unlikely to overcome the complete absence of skin perfusion, and thus should primarily serve an adjuvant role in combination with reversal therapy. HBOT should be implemented early and extended over multiple sessions during the first week post-injury, though treatment regimens vary [258].

Topical nitroglycerin (TNG) and oral phosphodiesterase inhibitors, initially conceived as important therapeutic components because of their vaso/venodilator properties, have yet to demonstrate a benefit in FIVO. TNG failed to show any significant benefit in case-control trials conducted in the rabbit ear model and may have a potentially deleterious congestive effect in affected tissues [111]. Anti-platelet therapy employing aspirin or clopidogrel, though lacking direct evidence of a therapeutic benefit in FIVO, represent a valid therapeutic intervention that is well-substantiated in other arterio-occlusive conditions aggravated by platelet activation and aggregation, such as myocardial infarction [259,260]. Warm compresses, recumbent positioning, and massaging—thought to aid in stimulating additional blood flow to the tissues or in filler dispersion—currently lack supportive evidence but are nonetheless advocated given their limited risk [86].

#### 3.5.2. Ischemic Cerebroretinal Injury (CRI)

In FIVO, intraluminal blockade of the cerebroretinal arterial tree represents the most damaging injurious event possible, unleashing a disabling and life-threatening cascade of neuronal tissue destruction. Cerebroretinal injuries (CRIs) arise by default from type IV filler disseminations, in which an occlusive plug retrogradely extends toward the origin of the retinal artery and back into the internal carotid/cerebral arteries. A recent systematic review of nearly 250 published instances of FIVO-associated facial skin injuries documented a 20% incidence of concomitant visual deficits and, of those, another 20% rate of ischemic stroke [25]. Patients with FIVO-associated cerebroretinal disturbances are more likely to present with dizziness/syncope upon injury. Inoculation of the ophthalmic cutaneous facial territory carries the highest risk of CRI and is responsible for >90% of such injuries. Nonetheless, because of the frequent occurrence of ECA-ICA anastomoses (Table 4)—which may traverse in superficial and deep planes—CRI may theoretically arise from injection in any region of the face [261].

Visual disturbances in FIVO may present with different combinations of blindness and ophthalmoplegia/ptosis, depending on the severity of occlusion of ciliary and retinal branches and the presence of communicating anastomoses (Table 5) [262]. Type IV injuries—consisting of blindness with ophthalmoplegia and blepharoptosis—are the most severe and common type of periocular complication associated with FIVO, with 90% of patients incurring some degree of blindness. Treatment of CRI has remained challenging because of the lack of prompt emergency therapeutic interventions for the management of this extremely time-sensitive condition. As previously discussed, intra-arterial therapy with combined thrombolytic/hyaluronidase delivery has shown the most promise when administered within 4.5 h post-injury [211]. HBOT may be beneficial in prolonging the survival time of retinal tissues until definitive therapy can be instituted in cases with partially intact posterior ciliary blood supply owing to the proximity of retinal tissues to the choroidal circulation [263].
molecules-27-05398-t004_Table 4Table 4Anastomoses between the internal and external carotid arterial systems described in the literature.AnastomosisReferenceDorsal nasal a. (ICA-OA)—Angular a. (ECA-FA)[264]Dorsal nasal a. (ICA-OA)—Infraorbital a. (ECA-IMA)[264]Supratrochlear a. (ICA-OA)—Anterior branch (ECA-STA)[264]Supraorbital a. (ICA-OA)—Anterior branch (ECA-STA)[264]Zygomaticofacial a. (ICA-OA)—Anterior branch (ECA-STA)[264]Zygomaticofacial a. (ICA-OA)—Transverse facial a. (ECA)[264]Zygomaticotemporal a. (ICA-OA)—Anterior branch (ECA-STA)[264]Lacrimal a. (ICA-OA)—Deep temporal a. (ECA-IMA)[264]Inferior palpebral a. (ICA-OA)—Infraorbital a. (ECA-IMA)[264]Anterior ethmoid a. (ICA-OA)—Sphenopalatine a (ECA-IMA)[264]Posterior ethmoid a. (ICA-OA)—Sphenopalatine a (ECA-IMA)[264]Meningo-ophthalmic a. (ICA-OA)—Middle meningeal a. (ECA-IMA)[264]Ophthalmic artery (ICA)—Artery of superior orbital fissure (ECA-IMA)[265]Petrous branch (ICA)—Middle meningeal a. (ECA-IMA)[266]Superior/tentorial branch (ICA-ILT)—Cavernous branches of MMA (ECA-IMA)[266]Anterolateral branch (ICA-ILT)—Orbital branches of MMA (ECA-IMA)[266]Anterolateral branch (ICA-ILT)—Artery of foramen rotundum (ECA-IMA)[266]Posteromedial branch (ICA-ILT)—Accessory meningeal a. (ECA-IMA)[266]Petrous ICA—Vidian a. (ECA-IMA)[266]ICA—internal carotid artery; ECA—external carotid artery; OA—ophthalmic artery; FA—facial artery; IMA—internal maxillary artery; STA—superficial temporal artery; ILT—inferolateral trunk; MMA—middle meningeal artery.

Cerebrovascular injuries may manifest with region-specific neurosymptomatology or present subclinically via incidental findings on radiological imaging [267,268,269,270]. The anterior and middle cerebral arteries and their territories, owing to their proximity to the ophthalmic artery, are the most commonly affected regions in FIVO; as such, patients with periocular complications should undergo immediate radiological screening for cerebral stroke. Patients with cerebral embolism typically present with altered consciousness, hemiplegia, headache, aphasia, and facial palsy. However, with a sufficiently large bolus, the entire bilateral anterior and posterior cerebral vasculature may be affected, posing life-threatening consequences. In a large review of all published instances of filler-induced cerebral embolism (FICE), Wang et al. documented the entire range of therapeutic interventions. The most commonly employed modalities included pharmacotherapy (steroids, antiplatelet agents, mannitol, and thrombolytics), HBOT, decompressive craniectomy, and intra-arterial mechanical/thrombolytic interventions. Despite treatment, the majority of patients with FICE suffered persistent functional deficits and 10% succumbed to injuries [26]. The early recognition and prompt initiation of therapy are critical to the successful management of CRI. Therefore, all injecting practitioners should be ready to perform a complete neuro-ophthalmological examination in all patients presenting with FIVO, regardless of the affected facial territory. Because numerous anastomotic connections exist between the internal and external carotid arterial systems, all facial cosmetic injections carry some risk of CRI. Practitioners are encouraged to proactively conduct internal preparedness assessments and establish emergency protocols that involve specialized care facilities in their region for prompt referral in cases of emergency.

## 4. Conclusions

The explosive growth in the utilization of hyaluronan-based dermal fillers in plastic surgery and aesthetic medicine has ushered a re-emergence of devastating sequelae stemming from inadvertent vascular occlusion. The mechanism of injury involves the accidental cannulation of a vessel followed by vaso-inoculation, vaso-dissemination, and subsequent arteriovascular blockade. The susceptibility of tissues to injury is directly influenced by the size and configuration of the accessible arteriovascular territory, the presence of collateral anastomotic perfusion, and the ischemic tolerance of affected tissues. The rheological properties of HA fillers, which can vary significantly between brands, may have direct implications on the type of intravascular dissemination that occurs as well as the rapid reversibility of the occlusion. Preventative strategies, such as minimization of bolus size, avoidance of high-risk regions, use of blunt-tipped microcannulas, and the adoption of US vascular mapping, may help decrease the risk and magnitude of potential injuries. Therapeutic interventions employing a combination of hyaluronidase with thrombolytic agents, administered via a variety of routes, hold promise in achieving the rapid recanalization of obstructed vessels. However, because of the limited survival time of ischemic cerebroretinal tissues, practitioners must remain vigilant to the early warning signs of vascular occlusion, ensuring the early recognition and prompt escalation of care to specialized emergency facilities. Patients presenting with symptoms of FIVO should receive a thorough neuro-ophthalmological examination to identify the presence of CRI. The FOEM scoring system and grading classification enhance the proper documentation and reporting of these rare, but significant events.

## Figures and Tables

**Figure 1 molecules-27-05398-f001:**
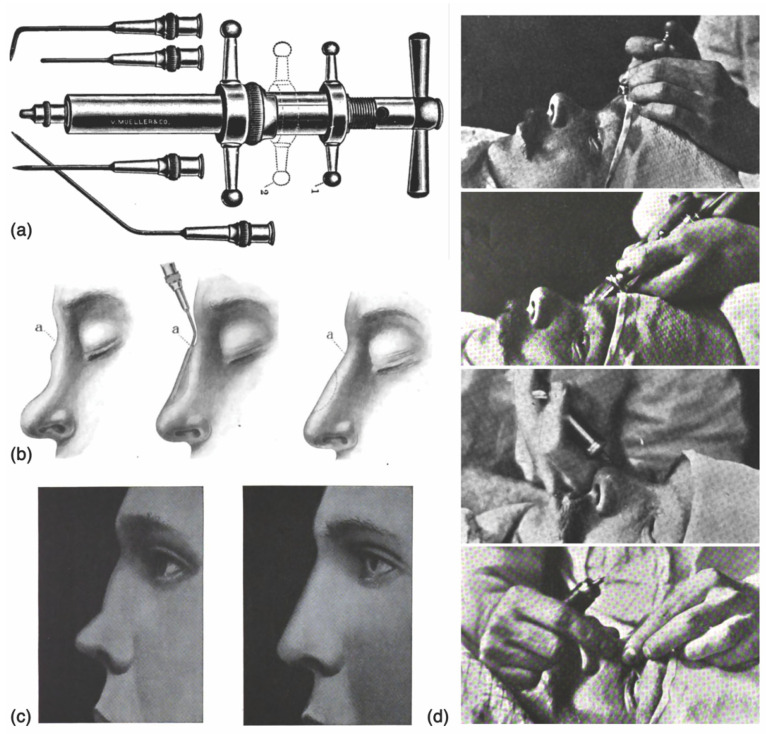
The history of cosmetic injectable fillers, early 1900s: (**a**) Paraffin syringe employed in paraffin facial injections, from Ballenger [6]. (**b**) Diagrammatic description of nasal dorsal augmentation with paraffin, ca. 1911, from Ballenger [6]. (**c**) Illustration of cosmetic patient before and following paraffin augmentation of the nasal dorsum, early ca. 1911, from Kolle [7]. (**d**) Procedural technique of paraffin injection in the correction of a dorsal nasal deformity, ca. 1908, from Miller [8].

**Figure 2 molecules-27-05398-f002:**
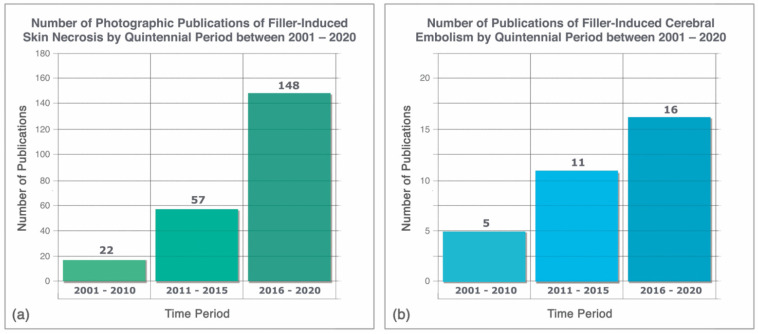
Increasing incidence in the number of published cases of filler-induced complications between 2001 and 2020: (**a**) skin necrosis, data from Soares et al. [25]; (**b**) cerebral embolism, data from Wang et al. [26].

**Figure 3 molecules-27-05398-f003:**
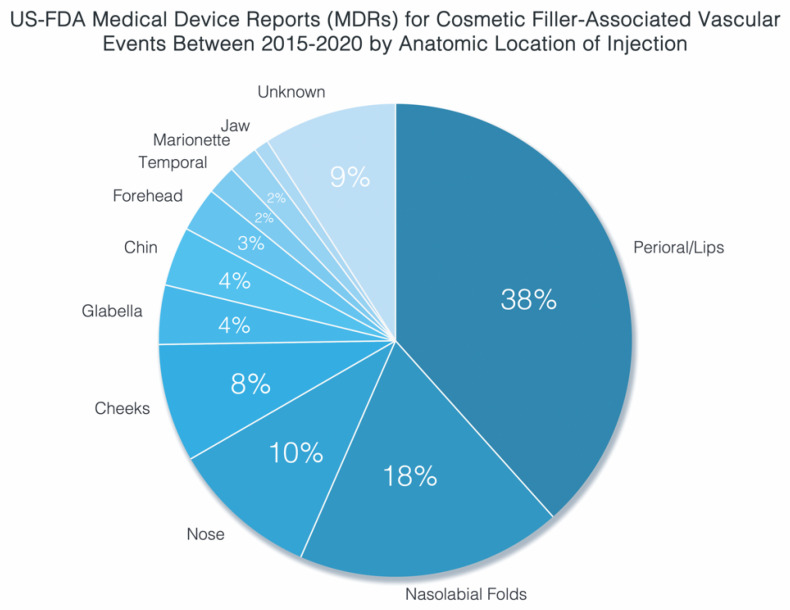
Food and Drug Administration (FDA) Medical Device Reports (MDRs) data for filler-associated vascular events reported between 2015 and 2020; data from [36].

**Figure 4 molecules-27-05398-f004:**
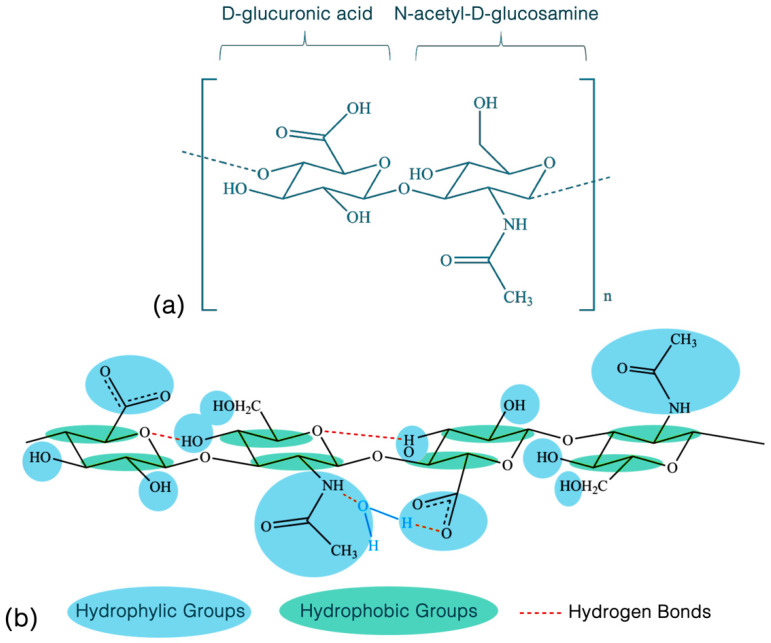
Chemical structure of the disaccharide unit of hyaluronan (HA). (**a**) Each HA molecule consists of variable-length chains made up of repeating disaccharide units, each composed of D-glucuronic acid and N-acetyl-D-glucosamine. (**b**) Each hyaluronan molecule features a large number of hydrophilic groups and intra-molecular hydrogen bonds, responsible for its high water-binding affinity and viscoelastic properties. Reproduced from Fallacara et al. [48], with permission.

**Figure 5 molecules-27-05398-f005:**
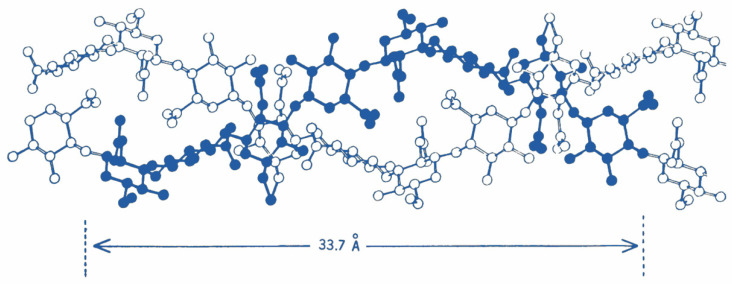
Hyaluronan molecules form secondary structures, including the proposed anti-parallel double helix shown here. These arrangements occur as a result of significant intra- and intermolecular hydrogen bonding, which influences the viscoelastic behavior of colloidal mixtures. Reproduced from Webber et al. [56], with permission.

**Figure 6 molecules-27-05398-f006:**
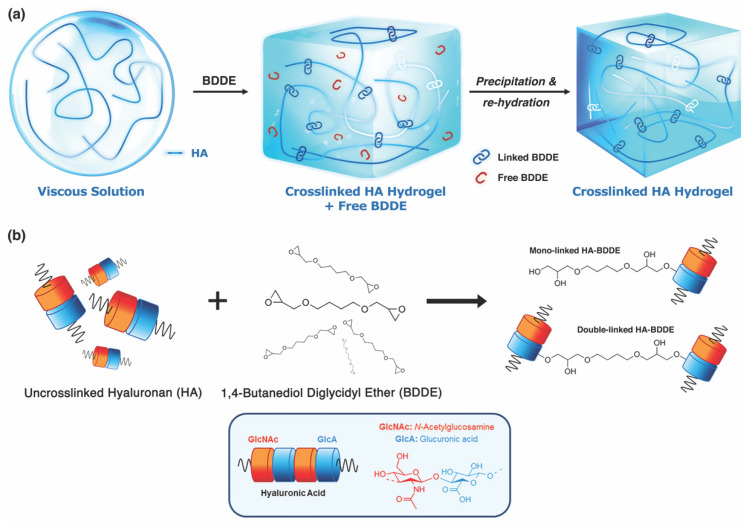
The synthesis and purification of crosslinked hyaluronan (HA) dermal filler hydrogels. (**a**) Uncrosslinked HA aqueous solution, a viscous fluid, is combined with 1,4-butanediol diglycidyl ether (BDDE), subsequently precipitated, and re-hydrated, yielding the finished product. The free, unlinked BDDE is eliminated from the final hydrogel. (**b**) Schematic representation of the molecular interaction of BDDE with HA molecules, yielding mono- and double-linked HA. Adapted from Guarise et al. [89], with permission.

**Figure 7 molecules-27-05398-f007:**
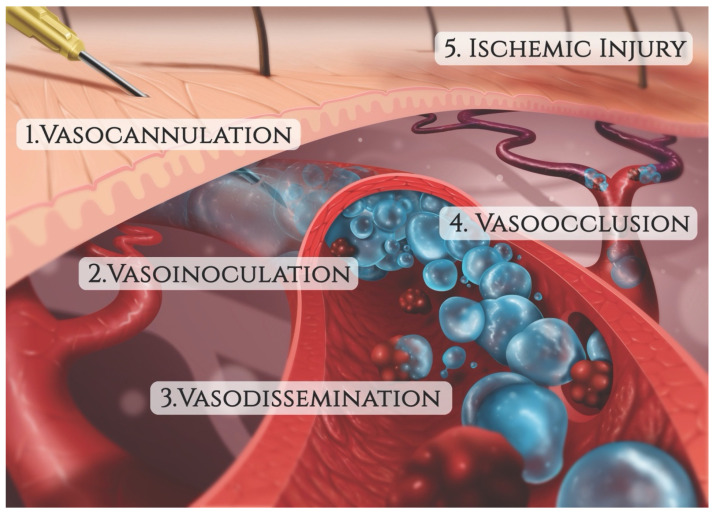
The sequence of events leading to filler-induced vascular occlusion (FIVO).

**Figure 8 molecules-27-05398-f008:**
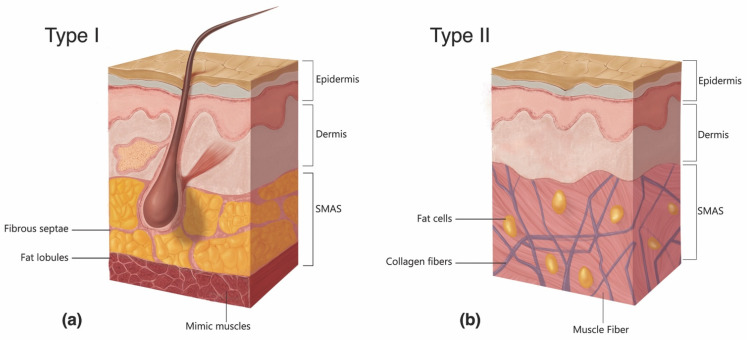
The two types of histomorphology of the facial superficial musculoapo-neurotic system (SMAS), according to Ghassemi et al. [127]: (**a**) Type I morphology is found predominantly in the cheek, forehead, and lateral face and features a uniform layer of adipose tissue interspersed amidst the fibrous septa, spanning the space between the SMAS and the dermis. (**b**) Type II morphology is present in the central facial tissues of the perioral and perinasal regions and is characterized by thick fibromuscular inser-tions emanating from the SMAS directly into the dermis, forming a more rigid adhesion between the two planes. Adapted from the classification by Ghassemi et al., *Aesthetic Plastic Surgery*, published by Springer Nature, 2003 [127], with permission.

**Figure 9 molecules-27-05398-f009:**
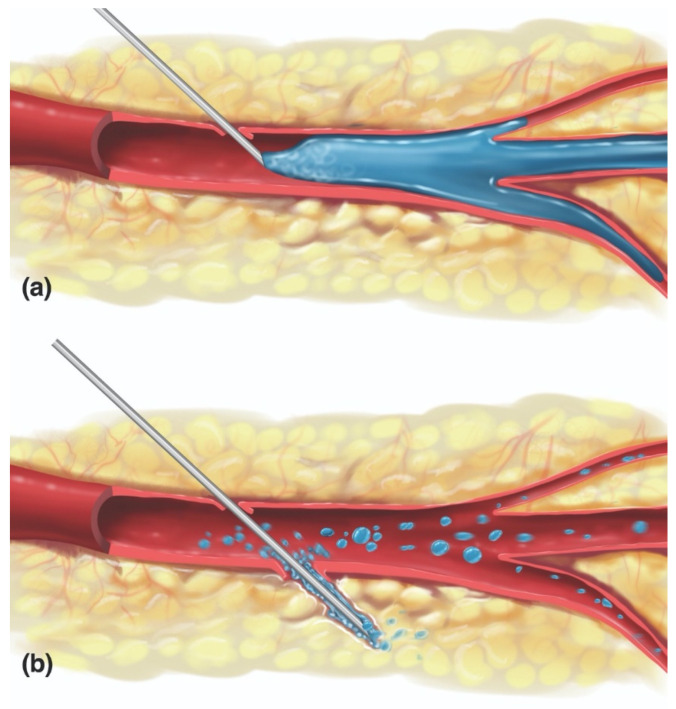
Mechanisms of vaso-inoculation. (**a**) Direct intraluminal injection. (**b**) Indirect extraluminal inoculation following trans-arterial perforation.

**Figure 10 molecules-27-05398-f010:**
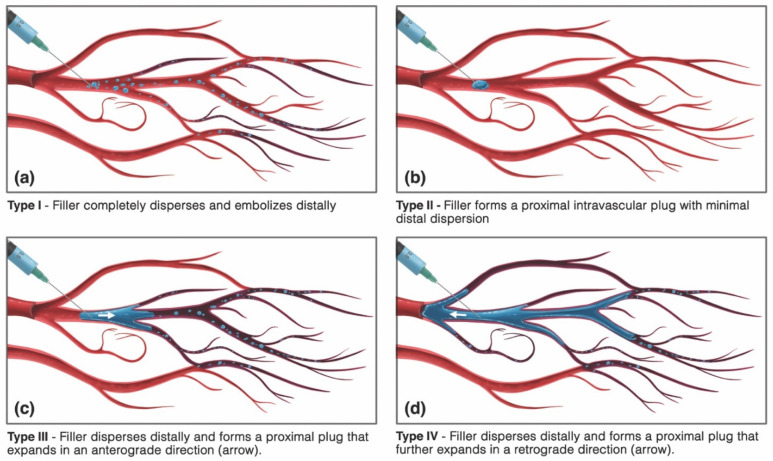
Possible vaso-dissemination mechanisms of intravascular hyaluronan gels. (**a**) Type I, filler completely disperses and embolizes distally, clearing the main lumen. (**b**) Type II, filler does not disperse and instead forms a proximally occlusive intraluminal plug. (**c**) Type III, filler disperses distally and forms a proximal intravascular plug that expands in an anterograde direction. (**d**) Type IV, filler disperses distally and forms a proximal intravascular plug that further expands in a retrograde direction, causing the additional occlusion of upstream branches.

**Figure 11 molecules-27-05398-f011:**
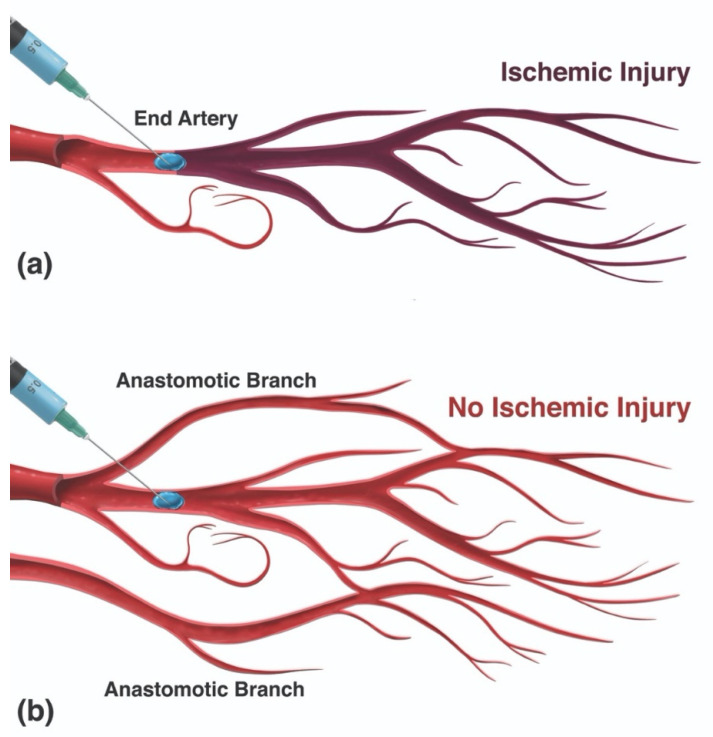
Influence of the configuration of a vascular network on the likelihood of tissue underperfusion/ischemia with a type II dissemination pattern. (**a**) In an end-arterial configuration, such as that seen with the retinal arterial system, a proximally obstructive plug can result in a significant decrease in tissue perfusion and a higher likelihood of ischemia. (**b**) In the presence of collateral perfusion supplied by anastomotic branches, a proximal plug is not likely to result in significant underperfusion or ischemia.

**Figure 12 molecules-27-05398-f012:**
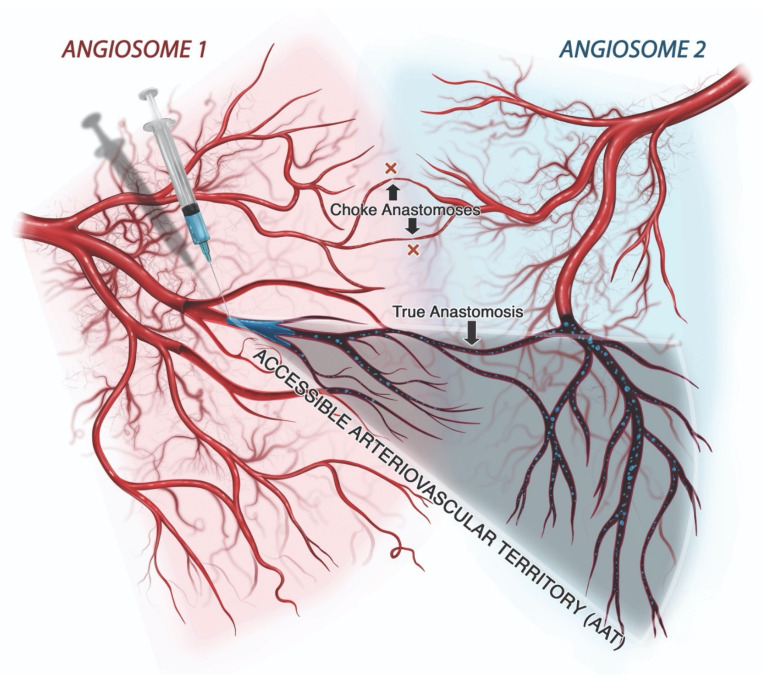
Conceptual illustration of the accessible arteriovascular territory (AAT). The AAT represents the conglomerate of vessels and their downstream anastomotic connections into which filler particles may disseminate following accidental intravascular inoculation. The AAT may involve more than one angiosome because of the presence of true anastomoses, which serve as vascular connections linking two angiosomes or, more distally, two perforasomes. Choke vessels, which function as gatekeepers, act as dynamic anastomoses that either permit cross-perfusion of tissues or, in cases of filler-induced vascular occlusion, limit the ability of filler to spread to adjacent angiosomes through a protective vasoconstrictive reflex.

**Figure 13 molecules-27-05398-f013:**
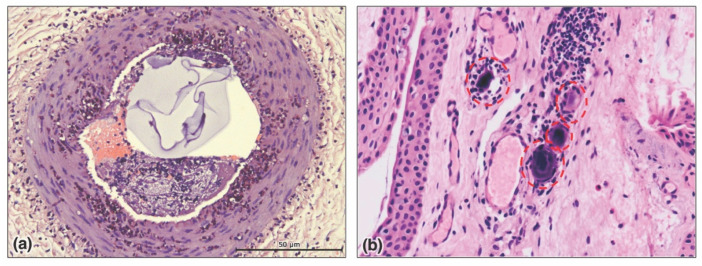
Histology of filler-induced vascular occlusion. (**a**) Hyaluronic acid (HA) dermal filler evident as an intraluminal plug within the central auricular artery in the rabbit ear model on post-injection day 1; from Zhuang et al. [166], with permission. (**b**) Histological section of bulbar conjunctiva of patient with filler-induced vision loss, showing occlusion of the distal microcirculation (capillaries and arterioles) with filler particles (red circles); from Hsiao et al. [176], with permission.

**Figure 14 molecules-27-05398-f014:**
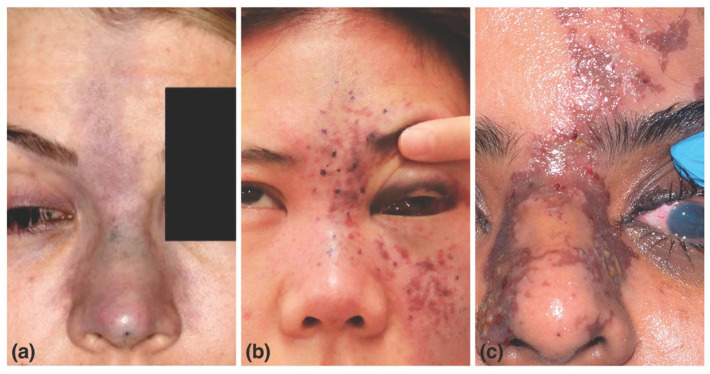
Diversity of ischemic skin patterns of the upper face following accidental vascular occlusion with hyaluronic acid (HA) gel filler. (**a**) Early ischemic changes (livedo reticularis) hours after dorsal nasal injection, showing involvement of the right supratrochlear territory, paracentral vessels, and dorsal nasal arterial regions; used with permission from Lucaciu et al. [187]. (**b**) Early ischemic changes 6 h following glabellar HA filler injection, demonstrating involvement of the left supratrochlear artery, paracentral vessels, dorsal nasal territories, and infraorbital region; used with permission from Xu et al. [188]. (**c**) Ischemic skin changes 24 h following accidental vascular occlusion resulting from nasal dorsal augmentation with HA filler, showing involvement of the left supratrochlear, left supraorbital, and bilateral dorsal nasal arteries; used with permission from Eldweik [193].

**Figure 15 molecules-27-05398-f015:**
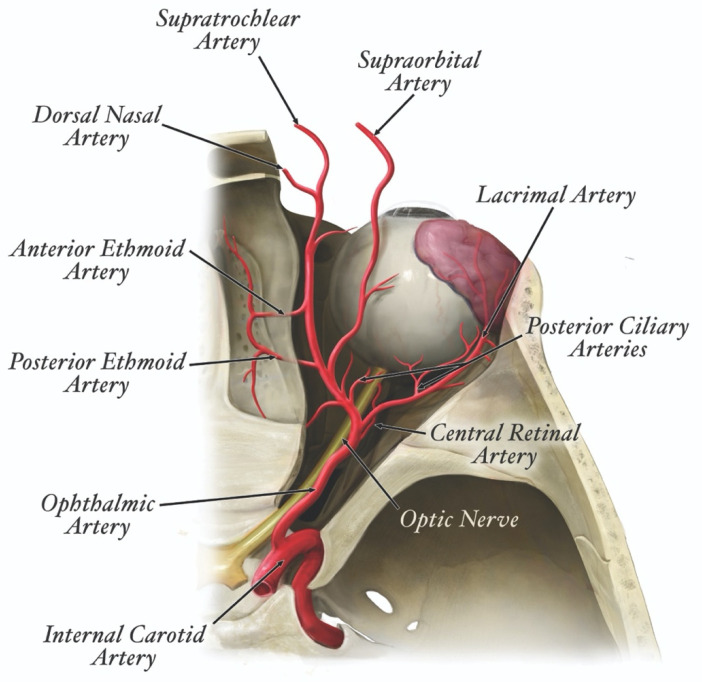
The vascular anatomy of the ophthalmic arterial system.

**Figure 16 molecules-27-05398-f016:**
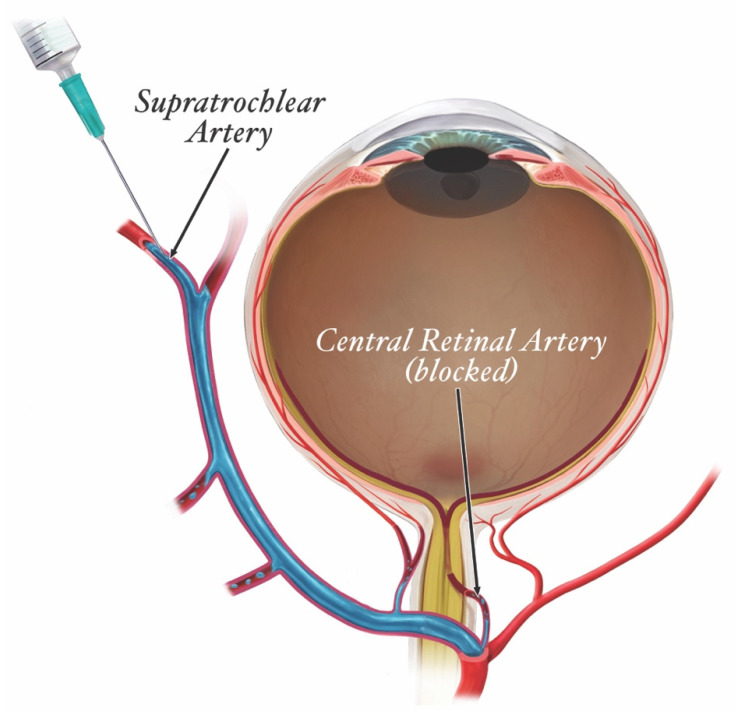
Mechanism of injury responsible for retinal ischemia and blindness from filler-induced vascular occlusion of peripheral origin. The ophthalmic arterial system ends in terminal cutaneous branches that supply the forehead, glabella, and nasal dorsum. Accidental cannulation of a distal cutaneous branch, such as the supratrochlear artery shown here, can result in a type IV dissemination of filler, creating a retrogradely expanding gel column that occludes upstream branches proximal to the site of cannulation. If the occlusive plug reaches sufficiently far to block the central retinal artery, interruption of this important end-arterial conduit will result in complete loss of perfusion to the macula and acute loss of vision.

**Figure 17 molecules-27-05398-f017:**
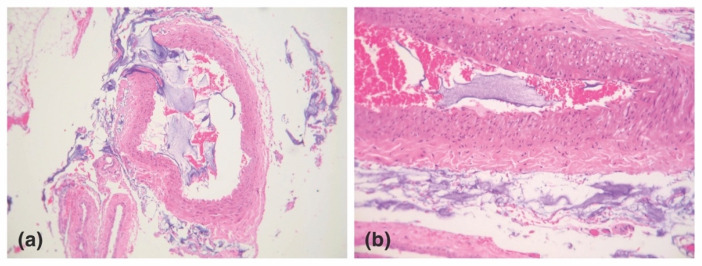
Histopathology of a hyaluronic acid (HA) filler-inoculated rat femoral artery demonstrating the thrombogenicity of HA gels. (**a**,**b**) Photomicrographs of hematoxylin and eosin (H&E) stained femoral artery sections at the site of arterial cannulation (**a**) and distally (**b**), showing the presence of intravascular thrombi associated with basophilic HA filler material, respectively; from Baley-Spindel et al. [175], with permission.

**Figure 18 molecules-27-05398-f018:**
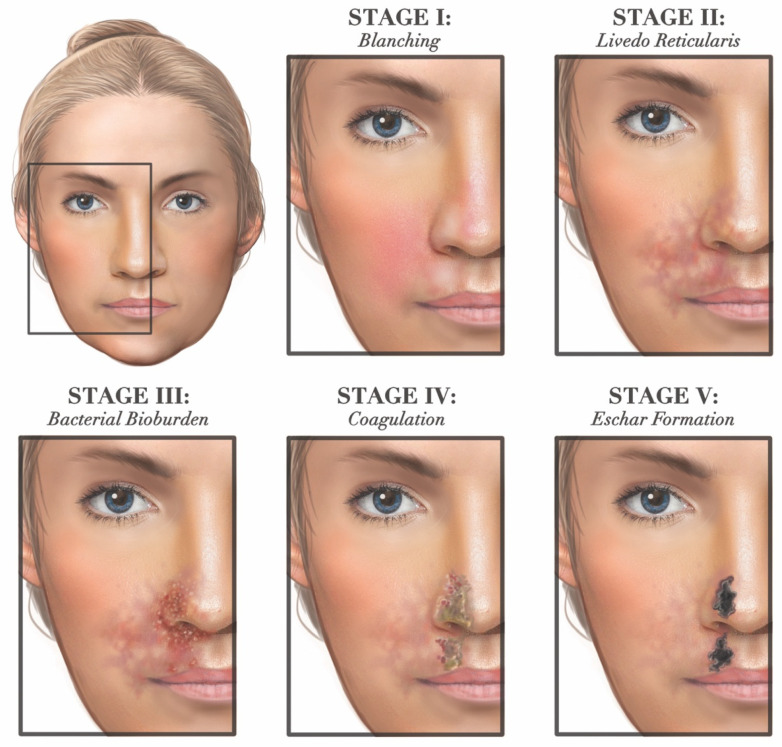
The clinical stages and appearance of the skin following an ischemic in-jury caused by vascular occlusion with dermal fillers, according to Murray et al. [86]: Stage I occurs immediately and is characterized by skin blanching with delayed capillary refill. Stage II manifests over 72 h, with livedoid skin changes due to dermal pooling of deoxygenated blood in underperfused regions. Stage III is characterized by the func-tional deterioration of the skin barrier with partial desquamation and overgrowth of cutaneous flora. Stage IV occurs 5–10 days post injury and is characterized by coagulative necrosis, with the eventual formation of an eschar (Stage V) which may persist for weeks. Adapted from the classification by Murray et al., *The Journal of Clinical and Aesthetic Dermatology*, published by Matrix Medical Communications, 2021 [86], with permission.

**Figure 19 molecules-27-05398-f019:**
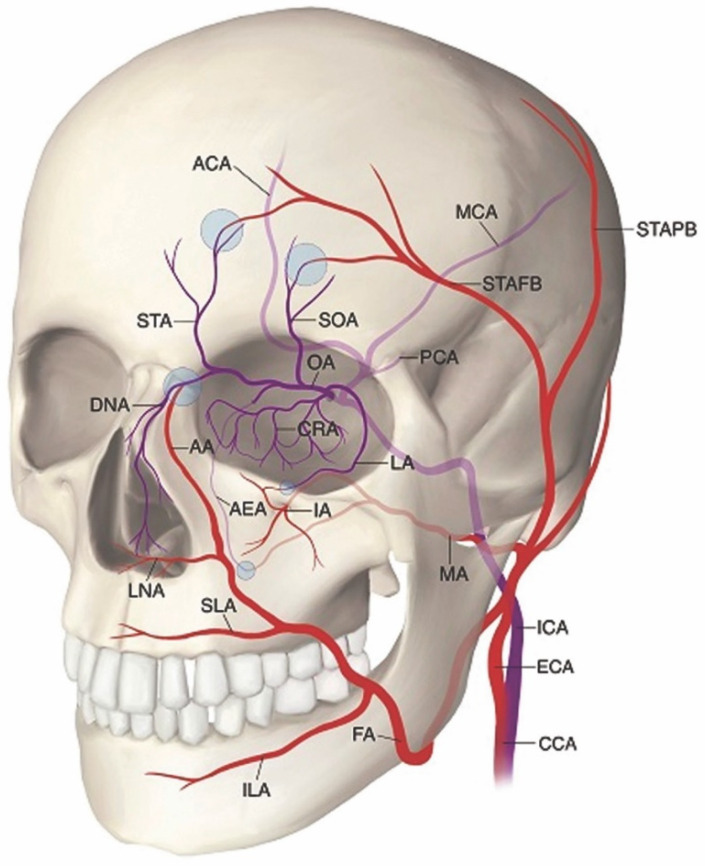
Schematic representation of the arterial vasculature of the head and neck comprised of the internal carotid (purple) and external carotid (red) main branches. The ophthalmic artery is a branch of the internal carotid artery and gives rise to terminal muco-cutaneous branches. These branches are involved in many of the anastomoses (blue circles) formed between the internal and external carotid arterial systems. AA, angular artery; ACA, anterior cerebral artery; AEA, anterior ethmoidal artery; CCA, common carotid artery; CRA, central retinal artery; DNA, dorsal nasal artery; ECA, external carotid artery; FA, facial artery; IA, infraorbital artery; ICA, internal carotid artery; ILA, inferior labial artery; LA, lacrimal artery; LNA, lateral nasal artery; MA, maxillary artery; MCA, middle cerebral artery; OA, ophthalmic artery; PCA, posterior cerebral artery; SLA, superior labial artery; SOA, supraorbital artery; STA, supratrochlear artery; STAFB, superficial temporal artery frontal branch; STAPB, superficial temporal artery parietal branch. From Wang et al. [26], with permission.

**Figure 20 molecules-27-05398-f020:**
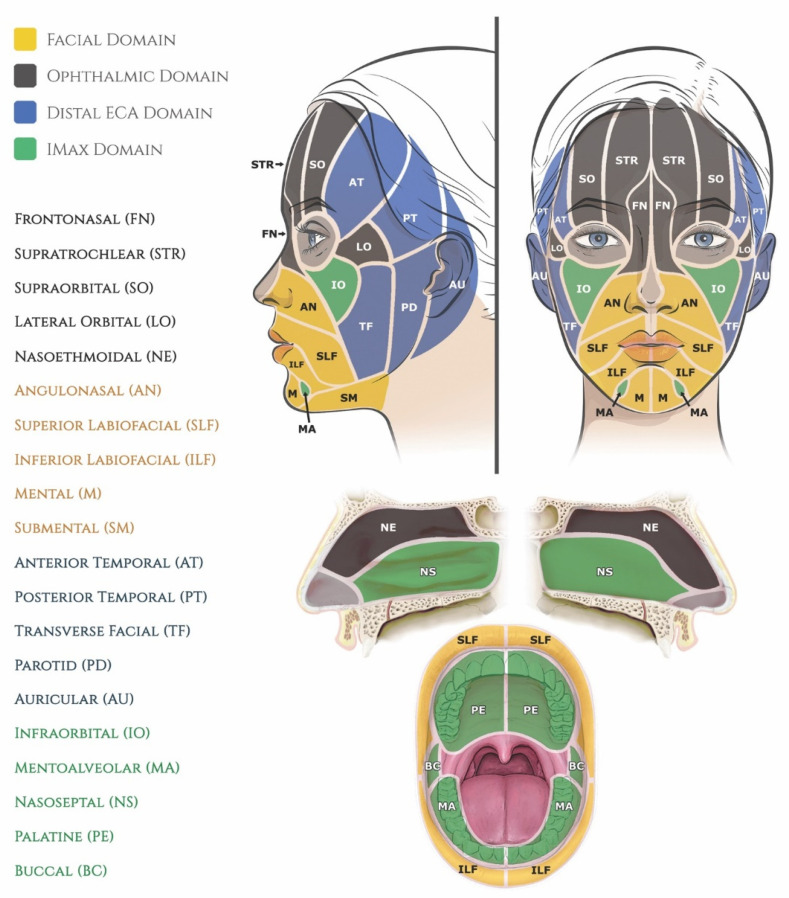
The facial, ophthalmic, distal external carotid, and internal maxillary (F.O.E.M.) domain angiosome scoring system for muco-cutaneous injuries secondary to filler-induced vascular occlusion. Each domain angiosome is subdivided into five mucocutaneous segments for a maximum bilateral score of 10 for each domain. From Soares et al. [29], with permission.

**Table 1 molecules-27-05398-t001:** Rheological properties of U.S. hyaluronic acid-based dermal filler brands. Data adapted from de la Guardia et al. [102].

Filler Product Name *	HA (mg/mL)	G’_5Hz_ (Pa)	G”_5Hz_ (Pa)	Tan δ	Cohesivity/Fn (gmf)	Swelling Factor (%)
**Belotero** *Balance*	22.5	128	82	0.641	69	664
**Juvéderm** *Ultra*	24	156	68	0.436	96	580
**Juvéderm** *Ultra XC*	24	207	80	0.386	96	622
**Juvéderm** *Ultra Plus*	24	214	74	0.346	116	515
**Juvéderm** *Ultra Plus XC*	24	263	79	0.300	112	454
**Juvéderm** *Volbella*	15	271	39	0.144	19	133
**Juvéderm** *Voluma*	20	398	41	0.103	40	227
**Restylane** *Refyne*	20	116	50	0.431	49	516
**Restylane** *Defyne*	20	342	47	0.137	60	318
**Restylane** *Kysse*	20	236	50	0.212	85	373
**Restylane**-*L*	20	864	185	0.214	29	<100
**Restylane** *Lyft*	20	977	198	0.203	32	<100
**Teosyal** *RHA1*	15	133	54	0.406	22	260
**Teosyal** *RHA2*	23	319	99	0.310	77	420
**Teosyal** *RHA3*	23	264	67	0.254	109	427
**Teosyal** *RHA4*	23	346	62	0.179	115	366

* All product trade names are the property of the respective owners (Belotero products, Merz Aesthetics; Juvéderm products, Allergan Aesthetics, an AbbVie company; Restylane products, Galderma Laboratories, LP; Teosyal products, Teoxane Laboratories). All products tested, except Juvéderm Ultra and Juvéderm Ultra Plus, contained lidocaine. HA—hyaluronic acid.

**Table 2 molecules-27-05398-t002:** Examples of proximal and distant arteriovenous occlusive sequelae that may arise from filler-induced vascular occlusion (FIVO) of facial-onset.

Arterial *	Venous
Muco-Cutaneous/Soft Tissue Necrosis	Local Venous Thrombophlebitis
Vision Loss +/− Ophthalmoplegia	Cerebral Sinus Thrombosis
Ischemic Cerebral Stroke	Pulmonary Embolism
Facial Paralysis/Peripheral Nerve Injury	Myocardial Infarction (PFO **)

* Arterial injuries may arise from venous inoculations due to the presence of arteriovenous shunts; ** Patent foramen ovale.

**Table 3 molecules-27-05398-t003:** The FOEM (facial, ophthalmic, distal external carotid, and internal maxillary) grading classification of mucocutaneous ischemic injuries. From Soares et al. [25].

**Grade I: Limited Mucocutaneous Injury**—No more than one FOEM Segment Affected
Ia: Ophthalmic domain not involved
Ib: Ophthalmic domain involved
**Grade II: Moderate Mucocutaneous Injury**—Two FOEM segments affected
IIa: Ophthalmic domain not involved
IIb: Ophthalmic domain involved
**Grade III: Extensive Mucocutaneous Injury**—Three or more FOEM segments affected
IIIa: Ophthalmic domain not involved
IIIb: Ophthalmic domain involved
**Grade IV: Visual Deficits**—Any segmental skin injury with visual deficits
IVa: Unilateral visual deficits
IVb: Bilateral visual deficits
**Grade V: Stroke**—Any segmental skin injury with ischemic stroke
Va: Unilateral ischemic stroke
Vb: Bilateral ischemic stroke

**Table 5 molecules-27-05398-t005:** Classification of blindness and periocular complications *.

Type 0	Ptosis and/or Ophthalmoplegia without Blindness
Type I	Blindness without Ophthalmoplegia and without Ptosis
Type II	Blindness with Ptosis but without Ophthalmoplegia
Type III	Blindness with Ophthalmoplegia but without Ptosis
Type IV	Blindness with Ophthalmoplegia and Ptosis

* Adapted from Myung et al. [262].

## Data Availability

Not applicable.

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
