# Peer review of "Bridging a Century-Old Problem: The Pathophysiology and Molecular Mechanisms of HA Filler-Induced Vascular Occlusion (FIVO)—Implications for Therapeutic Interventions"

_molecules, 2022, doi:10.3390/molecules27175398_

Round 1

Reviewer 1 Report

This paper is an well-organized review about the apthophysiology and molecular mechanisms of HA filler-induced vascular occlusion.

Author Response

Thank you for the positive feedback. 

Reviewer 2 Report

I would like to congratulate the authors. It is a very interesting review that honestly points out the possible complications of HA Fillers. It is written in great detail. Furthermore, the figures are also very illustrative. 

Author Response

Thank you for the positive feedback. 

Reviewer 3 Report

Dear Authors

Very interesting study.

Of course, the subject of the manuscript is very interesting. there are few reports in the literature that study the action of HA fillers at the vascular level and the possible complications.  The authors studied the phenomenon in a clear and correct way and the reporting of the results is honest and very interesting, potentially having an important impact on the future use of HA fillers.

congratulations!

Author Response

Thank you for the positive feedback.